# Melanocortins, Melanocortin Receptors and Multiple Sclerosis

**DOI:** 10.3390/brainsci7080104

**Published:** 2017-08-14

**Authors:** Robert P. Lisak, Joyce A. Benjamins

**Affiliations:** Department of Neurology, Wayne State University School of Medicine, Detroit, MI 48201, USA; jbenjami@med.wayne.edu

**Keywords:** ACTH, melanocortins, melanocortin receptors, multiple sclerosis, neuroprotection, oligodendroglia, repair

## Abstract

The melanocortins and their receptors have been extensively investigated for their roles in the hypothalamo-pituitary-adrenal axis, but to a lesser extent in immune cells and in the nervous system outside the hypothalamic axis. This review discusses corticosteroid dependent and independent effects of melanocortins on the peripheral immune system, central nervous system (CNS) effects mediated through neuronal regulation of immune system function, and direct effects on endogenous cells in the CNS. We have focused on the expression and function of melanocortin receptors in oligodendroglia (OL), the myelin producing cells of the CNS, with the goal of identifying new therapeutic approaches to decrease CNS damage in multiple sclerosis as well as to promote repair. It is clear that melanocortin signaling through their receptors in the CNS has potential for neuroprotection and repair in diseases like MS. Effects of melanocortins on the immune system by direct effects on the circulating cells (lymphocytes and monocytes) and by signaling through CNS cells in regions lacking a mature blood brain barrier are clear. However, additional studies are needed to develop highly effective MCR targeted therapies that directly affect endogenous cells of the CNS, particularly OL, their progenitors and neurons.

## 1. Introduction

Melanocortins and melanocortin receptors (MCR) have been extensively investigated for their roles in the hypothalamo-pituitary-adrenal axis [1,2,3,4,5], and to a lesser extent in immune cells [6,7,8,9,10] and the nervous system outside the hypothalamic axis [3,11,12]. We have focused on the expression and function of MCR in oligodendroglia (OL), the myelin producing cells of the central nervous system (CNS), with the goal of identifying new therapeutic approaches to decrease CNS damage in multiple sclerosis (MS) as well as to promote repair.

## 2. Melanocortins

The melanocortins—adrenocorticotropic hormone (ACTH), α-MSH, β-MSH and γ-MSH—are polypeptides derived from a common precursor, pro-opiomelanocortin (POMC) (Figure 1). ACTH is 39 amino acids in length, and can be cleaved to the smaller 13 amino acid α-MSH. The 12 amino acid γ-MSH is cleaved from the amino terminus of POMC, while the 20 amino acid β-MSH is cleaved from POMC towards the carboxy terminus. ACTH has both steroidogenic and nonsteroidogenic actions, while the other three melanocortins are nonsteroidogenic.

## 3. Melanocortin Receptors

Five subtypes of MCR have been identified and cloned; their distribution, function and pharmacology are characterized in part [1,2,4,13,14,15,16,17] (reviews). ACTH can activate all five receptor subtypes; of the melanocortins, only ACTH activates MC2R; ACTH as well as α-MSH, β-MSH and γ-MSH activate MC1R, MC3R, MC4R and MC5R, although with varying affinities. For example, α-MSH and β-MSH bind with higher affinity to MC1R, MC3R and MC4R than ACTH, while γ-MSH has relatively low binding affinities for all the MCR except MC3R [3,7].

All five MCR are G protein-coupled receptors of the Class A rhodopsin-like family and share 7 homologous membrane spanning domains but differ in their N-terminus and C-terminus sequences [13]. MC1R has been associated primarily with pigmentation, MC2R with glucocorticoid biosynthesis, MC3R and MC4R with energy homeostasis and MC5R with exocrine gland regulation [2,3,13]. However, each of the subtypes is widely distributed in various parts of the body, where they serve a variety of functions [2,3,6,7,12] (reviews).

Mutations in the genes of each of the MCR subtypes have been identified in humans and other mammals, as summarized by Switonski et al. [18]. MC1R is prominent in the synthesis of melanin in melanocytes, and mutations are associated with various skin phenotypes and diseases, including increased cancer risk, especially for melanomas [19,20,21]. Of interest, three reports indicate an association between disability in MS and MC1R gene single nucleotide polymorphisms leading to MC1R hyporesponsiveness [22,23,24]. Mutations in the human MC2R gene cause familial glucocorticoid deficiency; of the 25 missense mutations identified, most result in decreased trafficking of MC2R to the cell surface. A mutation in the MC2R gene with increased expression and stronger response to ACTH has been associated with increased responsiveness to ACTH treatment of infantile spasms [25,26]. MC3R, like MC4R, is involved in energy homeostasis; while several human mutations have been identified [20], there is a less clear association with obesity for these mutations than for MC4R mutations. MC4R has been extensively investigated in obesity research, and decreased activity of MC4R is the leading monogenic cause of severe early onset obesity [27,28]. Over 166 mutations in the human MC4R gene have been identified, many in obese individuals. MC5R is involved in lipid metabolism, exocrine function and inflammatory activity. Only a few polymorphisms have been identified in the human MC5R gene; associations with obesity, type 2 diabetes, schizophrenia and bipolar disorder have been reported [29,30].

## 4. Melanocortin Receptor Signaling

Melanocortin receptors are coupled to G proteins, and signal primarily via adenylyl cyclase and multiple down-stream pathways [31,32] (reviews). Other pathways independent of adenylyl cyclase and cAMP have also been identified. The signaling pathways are complex and pleiotropic, depending on the ligand, cell type, MCR surface expression, associated proteins, time of receptor occupancy and other factors. All 5 MCR are known to be coupled to Gs, but in some instances can be coupled to Gq or Gi [32] (review). MC1R activation through Gs stimulates adenylyl cyclase to increase cAMP with activation of protein kinase A (PKA), increases Ca^++^ levels, and can independently stimulate the ERK1/2 pathway, but appears to have little effect on the protein kinase C (PKC) pathway. In general, MC2R and MC3R signal in a similar fashion, but can also activate PKC. MC4R, in addition to coupling to Gs and activating adenylyl cyclase, can also couple to Gq or Gi to activate other signaling cascades, including PKC, PI3 kinase and ERK1/2 pathways. MC5R can independently activate the PKA pathway through Gs or the ERK1/2 pathway through Gi. G protein independent pathways involving a variety of kinases have been characterized for GPCRs; for example, MC1R activation via Src kinase has been reported [21]. Activation of MCR can also be regulated by receptor internalization; for all five MCR, binding of melanocortins or other agonists decreases association of β-arrestins, disrupting signaling via GPCRs and leading to internalization through clathrin-coated pits [32] (review). Regulation of MCR function occurs by multiple mechanisms, including transport to the plasma membrane by chaperones [33], attenuation and selectivity for signaling pathways via MCR membrane-associated proteins (MRAP) [34] and inhibition by the naturally occurring inhibitors agouti and agouti-related proteins.

In addition to melanocortins, the naturally occurring agonists of MCR, many pharmacologic agonists have been synthesized and tested for MCR subtype specificity, for longer half-lives than the rapidly degraded melanocortin peptides, and for reduction in steroidogenic or other side effects [4,11,35] (reviews). These include agents targeted to allosteric (extracellular), orthosteric (transmembrane, where ACTH and melanocortins bind) and signal transduction (intracellular) sites on MCR. The recent development of allosteric modulators [36,37] and biased agonists [4,38] present new approaches to activate MCR with potential therapeutic advantages. As one example, modulation of constitutively activated MC4R with inverse agonists has received recent attention as a promising area for drug development [39]. Conversely, naturally occurring antagonists include the proteins agouti and agouti-related protein. While development of specific antagonists for the ACTH-specific MC2R [40] (review) as well as for MC1R and MC5R has been problematic, synthetic antagonists are available for MC3R and MC4R [41]. The current development of cyclic peptides or site specific antibodies holds the promise of more subtype specific agents in the future.

## 5. Anti-Inflammatory Effects on the Peripheral Immune System

### 5.1. Corticosteroid Dependent Effects

Activation of MC2R expressed by cells of the adrenal cortex results in increase in circulating levels of corticosteroids. The control of corticosteroid production and secretion is under control of ACTH, which in turn is controlled by corticotropin releasing hormone (CRH) made by the hypothalamus. Levels of CRH and ACTH are controlled by levels of corticosteroids vain a feedback loop referred to as the pituitary-adrenal axis. Administration of ACTH, the only melanocortin with the ability to strongly bind to MC2R [2,4,6] and to initiate signaling of MC2R, results in an increase of corticosteroids. The corticosteroids have extensive effects on many body functions including exerting anti-inflammatory effects on many cells of the immune system. The effects of corticosteroids on the trafficking, number and function of the lymphocytes and monocytes have been assumed to be the mechanism of therapeutic efficacy of ACTH in MS and other immune/inflammatory mediated diseases. As noted in Section 6, there is evidence that effects of ACTH and other melanocortins on immune and other inflammatory processes may also involve direct effects on circulating cells of the immune system, effects on the immune system via the CNS and effects on cells within the CNS [4,6,8,9,42].

### 5.2. Corticosteroid Independent Effects

Inflammatory cells including lymphocytes, monocytes**/**macrophages, and neutrophils as well as tissue-based cells including mast cells express MCR. Monocyte/macrophages as well as microglia express MC1R, MC3R and MC5R and lymphocytes express MC1R, MC3R and MC5R as well, reviewed in [6]. Signaling through these receptors inhibits inflammatory processes and has been associated with shifts from proinflammatory to inhibitory effects of lymphocytes, perhaps in part through effects on antigen presenting cells such as monocytes/macrophages. While this is somewhat difficult to demonstrate with administration of ACTH which increases endogenous corticosteroids, studies with α-MSH, which does not signal through MC2R, show induction of anti-inflammatory effects [43,44,45,46]. Administration of intravenous ACTH is able to inhibit maturation of B cells obtained from those MS patients into immunoglobulin secreting cells in vitro [47]. α-MSH inhibits inflammation in experimental autoimmune uveitis (EAU), induces CD25+ regulatory CD4+ T-cells [45,46,48] and regulates ubiquitination in T cells as well [49]. MC5R appears to be the important MCR in these latter functions. Nonsteroidogenic effects of ACTH were demonstrated in a rat model of gouty arthritis; ACTH administered systemically did not reduce joint inflammation, whereas ACTH or the MC3R agonist γ-MSH injected locally reduced inflammation in both normal and adrenalectomized rats [50]. Additional studies are clearly needed to study longer term effects of ACTH treatment in patients as well as in animal models to determine if there are long-term effects that outlast any immediate effects of the increase in corticosteroids on the peripheral immune system.

## 6. Direct Effects in the CNS

### 6.1. Effects Mediated through Neuronal Regulation of Immune System Function

Endogenous ACTH and other melanocortins, as well as presumably exogenously administered ACTH, can access the CNS in the brain stem and the hypothalamus, bind to MCR, particularly MC4R, and initiate signaling [10,51,52,53,54,55,56,57,58]. These brain stem neurons trigger vagal activity with release of acetylcholine (ACh) in peripheral tissue, with binding and activation of acetylcholine receptors (AChR). An important receptor is α7-AChR, which triggers anti-inflammatory processes and inhibition of excitatory and other damaging processes [59]. It has been suggested that some of this anti-inflammatory activity may then feed back to the CNS and further provide neuroprotective effects within the CNS [6,9,60] (reviews). Bilateral vagotomy interferes with melanocortin protective effects in the CNS [58], supporting the importance of the vagal pathway in melanocortin neuroprotection and reparative processes [51,57]. Activation of hypothalamic neurons via MCR seems to be important in hormonal and metabolic processes as noted in Section 3 [11,13,61]. These important pathways are covered in greater detail in several review articles [4,6,8,9,11].

### 6.2. Effects on CNS Neurons

Within the CNS, neuronal expression of MCR has been characterized most extensively in the central hypothalamic melanocortin pathway, where MC4Rs are the subtype involved in regulation of metabolism [5] (review). For example, genetic regulation of MC4R expression in cholinergic neurons in several extrahypothalamic brain regions implicate MC4R in regulation of energy balance and glucose homeostasis [28,62]. A recent study shows that constitutive activity of MC4R inhibits L-type voltage-gated calcium channels in cultured neurons [63]. Activation of MC4R shows neuroprotective and neuroregenerative effects in several models of neurodegenerative diseases [52], including neurogenesis and cognitive recovery in an animal model of Alzheimer’s disease [64]. We reported that the MCR agonist ACTH1-39 protects cultured rat forebrain neurons from excitotoxic, apoptotic, oxidative and inflammation related insults [65], but the specific MCR subtypes involved are not known.

Melanocortins are increasingly being investigated for their effects on synaptic remodeling [3] (review). For example, in the hippocampal C1 region, activation of MC4R at the postsynaptic ending increases cAMP levels and activates PKA, thus modulating long-term potentiation and long-term depression. In dopaminergic neurons, cross talk between MC4R and dopamine receptors regulates increased expression of AMPA receptors to increase dopamine responsiveness, or promotes endocytosis of AMPA receptors to reduce long-term depression [3].

### 6.3. Effects on Glia

MCR expression and function has been previously characterized in astroglia, and to a lesser extent in microglia and Schwann cells. Our recent studies, summarized in later Section 7 and Section 8 of this review, are the first to examine MCR expression and subtypes in OL and their precursors. Melanocortin effects on astroglia point to their roles in inflammation, obesity and regeneration [66] (review). Experiments to date in astroglia indicate that message for MC1R and MC4R but not MC3R is expressed, as analyzed by RT-PCR [7,67]. An early study on astroglia with a panel of MCR agonists showed that morphologic changes, including rounding of the cell body and process extension, were mediated by a cAMP mediated pathway, while proliferation was stimulated by an alternative pathway independent of cAMP [68]. More recently, MC4R activation in astroglia with the long acting α-MSH analogue NDP-MSH was shown to increase expression of brain-derived neuronotrophic factor [69] and stimulate the release of the anti-inflammatory TGF-β [70], in part via the ERK-cFos pathway. Microglia, the endogenous macrophages of the CNS, express MC4R [70] as well as MC1R, MC3R and MC5R [7,8]. Melanocortin peptides decrease microglial production of nitric oxide (NO) and the proinflammatory cytokines TNF-α and IL-6 [43,71], but increase release of the anti-inflammatory cytokine IL-10 [70]. In addition, NDP-MSH promotes an M2-like phenotype in microglia and inhibits microglial activation induced by Toll-Like Receptors 2 and 4 [72]. MCR expression and function have been less well studied in Schwann cells in the PNS. ACTH promotes peripheral nerve regeneration and axonal growth in vivo [73], while α-MSH inhibits inflammatory signaling in cultured Schwann cells [74]. The melanocortin analogue Org2766 as well as α-MSH stimulates Schwann cell proliferation, upregulates the NGF low-affinity receptor p75, induces release of an unidentified neurotrophic activity [75] and enhances nerve regeneration [76].

## 7. Effects on Endogenous Cells of the CNS with Potential Protective and Reparative Importance in MS and Other CNS Disorders

As noted in Section 6, MCR are expressed by neurons, astroglia, microglia and OL with some differential expression in different regions of the brain. Additionally there is in vitro and in vivo evidence, cited earlier, that these receptors are functional. Since MCR are known to be present within the CNS, some of the therapeutic effects in EAE and in the PNS experimental model experimental autoimmune neuritis (EAN) as well as in other animal models and in human diseases might be independent of the stimulation of endogenous corticosteroid production via MC2R signaling in the adrenal gland. While some of these non-corticosteroid disease modifying and anti-inflammatory effects might be due to stimulation of MC1R and other MCR expressed by peripheral immune cells, including lymphocytes and monocytes, direct effects on endogenous CNS cells may also occur.

An important animal model of MS is experimental autoimmune encephalomyelitis (EAE, originally called experimental allergic encephalomyelitis), which is induced by sensitization of experimental animals with CNS tissue, CNS myelin or specific constituents of myelin including myelin basic protein (MBP, originally called basic protein or encephalitogenic protein), proteolipid protein (PLP) and myelin oligodendrocyte glycoprotein (MOG). Alternatively, EAE can be induced in naive animals by passive transfer of lymphocytes, T cells or T cell lines or clones. Depending on the sensitizing antigen, the sensitization protocols, the species and strain of animals employed acute, hyperacute, chronic and relapsing courses of EAE can develop. Prevention and treatment of EAE can be achieved with many agents and treatments with EAE serving as a test system to screen for potential treatments for MS, both relapses and as disease modifying therapies [77,78]. ACTH was among the first experimental therapy employed in EAE [79] and as treatment for relapses (exacerbations) of MS; see Section 9, Treatment of Neurologic Diseases with Melanocortins.

Inhibition of EAE by ACTH may involve both corticosteroid and non-corticosteroid effects on the immune system. As noted there is indirect evidence of protective effects within the CNS including less demyelination along with evidence of repair; i.e., remyelination [80]. α-MSH, which cannot signal through MC2R expressed in the adrenals, inhibits the development of EAE [81,82] and EAN [83], a peripheral neuropathy, that serves as a model for some variants of Guillain-Barre Syndrome (GBS). An α-MSH analog, SValpha-MSH also inhibits EAE, acting to inhibit CD4+ T cells [84]. Since α-MSH cannot increase endogenous corticosteroids and yet inhibits development of EAN, this supports the idea that inhibition of EAN is due to the direct effects of α-MSH on immune cells and/or Schwann cells, the myelin forming cells of the PNS. Indeed it has been shown that a-MSH inhibits the translocation of NFκB in Schwann cells in vitro [74] demonstrating the potential for a direct effect of ACTH on Schwann cells in ameliorating EAN. Similarly, NDP-MSH, a long-lived analog of α-MSH, ameliorates EAE and restores BBB in mice; in vitro, protection of mouse and human neurons from excitotoxicity occurred via MC1R activation [85]. Melanocortins have been reported to be neuroprotective in animal models of excitotoxic injury [86], subarachnoid hemorrhage [87], traumatic CNS injury [88], an animal model of Alzheimer’s disease [64] and in peripheral nerve injury and repair [76,89,90] as well as the previously mentioned EAN and EAE.

Melanocortins have been shown to provide protection in vitro for neurons from toxic effects of cisplatin [91], protect neuronal cell lines from serum-induced apoptosis [92], provide trophic effects to neurons [93] and enhance neurite outgrowth in vitro [94]. Melanocortins can inhibit production of proinflammatory molecules by microglia [71]. Prior to our investigations, there has been little work done on MCR expression and effects of melanocortins on OL function. MCR expression in vitro has been discussed earlier and importantly all MCR are expressed by OL (Figure 2) [95]. The expression of MC4R in differentiated OL is shown in Figure 2A.

In order to dissect the mechanisms that may be involved in interactions between ACTH and other melanocortins and cells of the OL lineage, we undertook a series of experiments employing glial and neuronal cultures. ACTH inhibits death of OL and OPC induced by several mechanisms that are involved in damage to the CNS in MS as well as other disorders of the CNS, including glutamate (excitotoxicity) [96,97], apoptosis (induced by staurosporine, a widely employed molecule in apoptosis research) [96,97], reactive oxygen species (ROS) induced by hydrogen peroxide (H_2_O_2_) [96,97] and inflammation mediated by quinolinic acid (QA), a product downstream of kynurenic acid in the tryptophan indoleamine pathway [96,97] (Table 1). In the case of glutamate induced OL and OPC death, ACTH inhibited cell death mediated through all three of the ionotropic glutamate channels, NMDA, AMPA and kainate [96,97]. There was no protection of OL and OPC from toxicity induced by kynurenic acid, an earlier metabolite in the indoleamine tryptophan inflammatory pathway [96,97]. ACTH did not protect OL from either slow or rapid release of nitric oxide (NO) but provided modest protection of OPC from slow but not rapid release of NO [96,97].

Neurons and axons are also targets of pathologic processes in MS and therefore we also examined whether ACTH could protect neurons from these same pathologic mechanisms. As noted, neurons in several regions of the CNS are known to express MCR. ACTH inhibited neuronal death induced by staurosporine, quinolinic acid and ROS induced by H_2_O_2_ as well as glutamate, including via NMDA, AMPA and kainate (Table 1). ACTH protected neurons from death induced by rapid release of NO but not slow release, the opposite of the findings in OPC. As with OL and OPC, ACTH failed to provide any protection from cell death induced by kynurenic acid [65].

In order to determine whether protection of OL was a result of direct effects of ACTH on OL, or whether astrocytes or microglia, which express MCR and are present in our mixed glial cultures, were potentially involved in protection of OL from the different toxic molecules, we undertook another series of experiments. Using highly purified OL cultures, we found that ACTH directly protected OL from staurosporine (apoptosis), H_2_O_2_ (ROS), glutamate including NMDA, AMPA and kainate, and quinolinic acid. As with the mixed glial cell cultures, OL in purified cultures were not protected from kynurenic acid or NO. Conditioned medium from astrocytes treated with ACTH was able to protect purified OL from glutamate, AMPA, quinolinic acid, and ROS but not from kainate, staurosporine, kynurenic acid or NO [98]. Thus, astrocytes may contribute to protection of OL from some but not all toxic molecules. Similar experiments with conditioned medium from microglia treated with ACTH failed to provide protection, suggesting that if microglia are also providing help in protecting OL from these molecules it is not through secreted molecules, but potentially could occur through cell-cell interactions.

In addition to MCR being important in inhibiting inflammation via stimulation of endogenous corticosteroid production, acting directly on cells of the immune system and providing protection for OL, OPC and neurons, we have been interested in the potential of MCR signaling to contribute to repair in the CNS in MS, as well as in other diseases of the CNS. Incubation of mixed glial cell cultures, containing mature OL, with ACTH resulted in striking extension of the OL membrane [97] (Figure 2). Incubation of OPC with ACTH resulted in both an increase in the rate of OPC proliferation and rate of maturation from OPC (expressing platelet derived growth factor alpha receptor; PDGFαR) to cells expressing both PDGFαR and O1, a marker of galactolipids (a phenotypic marker of mature OL) and to cells expressing only O1, i.e., mature OL. Since the OPC themselves, not being a clone but rather primary cultures, may be at different stages of maturation, this likely explains the effect of both enhanced OPC proliferation and maturation. An increase in the number of OPC and more rapid maturation of OPC into mature OL both have the potential to enhance repair by increasing remyelination.

## 8. Melanocortin Receptor Signaling in Oligodendroglial Protection

As described earlier, we reported that OL express MC4R in vitro and more recently, we have shown that OL also express MC1R, MC3R and MC5R but not MC2R [95]. Employing agonists and antagonists, we have found that MC1R, MC3R, MC4R and MC5R are functional in protection of OL from the toxic effects of staurosporine, glutamate, quinolinic acid and ROS. We have also shown that MC4R is functional in protecting OPC and stimulating their proliferation, but have not yet investigated the function of the other MCR subtypes in OPC. Additional studies employing other strategies including silencing RNA will be required to further characterize the relative roles of each of these receptors in signaling for protection as well as in OPC proliferation and maturation.

To further understand the role of MCR in protection of OL from cytotoxic mechanisms important in the pathogenesis of the MS lesion, we examined different signaling pathways activated by ACTH, which is known to bind and signal via all 5 MCR. To do this, we tested the ability of inhibitors of several intracellular signaling pathways to block the ACTH inhibition of toxicity of the different cytotoxic molecules. Purified OL cultures were incubated with inhibitors of PI3 kinase, MAP kinase (MAPK) and protein kinase C α,β isoforms (PKCα,β) followed by ACTH and the test molecules with the controls of the toxic molecules and toxic molecules and ACTH without prior incubation with the kinase pathway inhibitors [65,99]. PI3 kinase is used for ACTH protection from staurosporine (apoptosis), quinolinic acid (inflammation), glutamate including NMDA, AMPA and kainate (excitotoxicity). The MAP kinase pathway was also used for protection from staurosporine, and glutamate. Neither pathway was involved in ACTH induced protection from ROS. MCR are known to signal by activating adenylyl cyclase and upregulating intracellular cyclic adenosine monophosphate (cAMP). Inhibition of adenylyl cyclase prevented ACTH from protecting OL from the toxic effects of glutamate, quinolinic acid, ROS and staurosporine, demonstrating that in our system ACTH signaling involves activation of adenylyl cyclase [99]. PKCα,β inhibition did not block or enhance ACTH protection from any of the toxic molecules but inhibition of PKCα,β *per se* protected OL from the same molecules as ACTH, suggesting that cell death from those molecules involves the PKCα,β pathway or alternatively that inhibiting PKCα,β activates adenylyl cyclase.

## 9. Treatment of Human Neurologic Diseases with Melanocortins

ACTH has been used as treatment for a wide variety of non-neurological diseases including nephrotic syndrome, sarcoidosis, and rheumatologic disorders [4], but has been less explored for treatment of neurologic diseases. ACTH in a depo form is called ACTHar gel. It is prepared from pituitary extract and likely contains other peptides and melanocortins, including α-MSH as a breakdown product of ACTH. ACTH is used as treatment for West syndrome, which is characterized by infantile spasms and an EEG pattern referred to as hypsyrrythmia. In several studies, ACTH has been found to be more effective than corticosteroids [100,101,102], suggesting that ACTH may act, in part, independent of the ability to increase levels of endogenous corticosteroids. Studies have suggested that exogenous ACTH does not readily cross the blood CSF barrier but this may be different than the blood brain barrier (BBB), and CSF levels are what have been examined in patients; see Section 10, Future Directions. Endogenous ACTH appears to be lower in the CSF of patients with West syndrome but treatment with ACTH does not seem to increase concentrations of total CSF ACTH [103,104,105]. ACTH also stimulates production of the mineralocorticoid deoxycorticosterone by the adrenal cortex. This molecule can be metabolized to allotetrahyrodeoxycorticostereone, a neurosteroid that is known to cross the BBB [106]. In the case of activating brain stem neurons, the BBB is absent in parts of the brain stem. Direct effects on abnormally firing cortical neurons may be possible, since ACTH and other melanocortins are small polypeptides and the BBB is not fully developed in infants. There is also a report on higher levels of CSF corticosteroids in patients with opsoclonus myoclonus treated with ACTH than treated with corticosteroids [107].

ACTH is used for the treatment of relapses of MS and was the first agent found to be effective in shortening the duration of relapses [108,109,110,111,112,113]. It is now administered intramuscularly as ACTHar gel. ACTH has been mainly replaced by very high doses of corticosteroids administered intravenously or by mouth, resulting in much higher but shorter lived blood levels of corticosteroids when compared to blood levels resulting from ACTH [114]. In one head to head study, ACTH has been shown to be equally effective in reducing duration of relapses compared to corticosteroids [113]. In a small study, dexamethasone was superior to methylprednisolone and ACTH in shortening duration of relapses, but there were only 30 patients in that study [115]. More recently, a small randomized open label rater blinded study demonstrated that ACTH was more effective than intravenous methylprednisolone for relapses and had greater effects on plasma cytokines, when added to interferon beta [116]. ACTH as treatment for relapses is generally reserved for patients who are allergic to corticosteroids, develop psychosis with corticosteroid therapy or who fail to respond to treatment with corticosteroids. Whether the beneficial effect on relapses is due to corticosteroids, direct effects of ACTH on immune cells and/or effects on endogenous cells of the CNS is not clear and may well involve all of these mechanisms. There are no studies on ACTH entry into the CNS in any animal models but again ACTH and other melanocortins are relatively small molecules and in relapses it is clear that large proteins, including serum albumin and immunoglobulins (Ig) enter the CNS. ACTH and ACTH followed by prednisone were more effective in reducing CSF IgG synthesis rate than oral prednisone alone, dexamethasone or intrathecal hydrocortisone. However, oligoclonal bands persisted and there was no clinical effect in a group of patients who were in progressive stage of MS [117]. What is needed is a large study comparing the longer term effects of ACTH with corticosteroids for relapses using both clinical outcomes as well as MRI, VEP and OCT to determine if the use of ACTH, which has both steroidogenic and non-steroidogenic effects, is superior to corticosteroids for treatment of relapses. A study of secondary progressive MS (SPMS), for which there is currently no approved therapy other than the mitoxantrone with the problems of sterility, congestive heart failure and leukemia, would also be of interest, again using imaging as one of the outcomes. There are currently a large number of studies underway testing the effect of ACTH activation of MCR in different diseases including several neurologic disorders [4].

## 10. Future Studies

There is a need to further characterize the relative roles of MC1R, MC3R, MC4R and MC5R in vitro and in vivo so that one might develop new small molecules that bind with high avidity to the specific receptors most important for protection and repair. Silencing genes specific for each receptor and testing both the in vitro and in vivo effects would be a useful approach to this problem.

While there are many studies on the effects of ACTH on EAE, more studies are needed examining the effects on passive EAE and chronic EAE; the former to look at effects in MCR signaling limited to the effector stage of EAE and the latter to see if there are protective and reparative effects during a more chronic stage of EAE at a time the inflammatory phase of the disease with influx of cells from the peripheral immune system into the CNS is less marked. Chronic EAE is a better model for progressive phases of MS than acute EAE. In addition, studying the effects of natural melanocortins like ACTH and α-MSH on non-immune mediated models of demyelination including acute and chronic cuprizone poisoning and CNS lysolecithin injections would allow one to separate demyelination and remyelination from the role of exogenous inflammatory cells as in these models where the inflammation is basically activation of the endogenous microglia.

Studies using the available ACTH as ACTHar gel looking the effects at one and two years post-treatment of a relapse in comparison to corticosteroids employing advanced MRI metrics, MRS and/or OCT in a phase 2 study would be of interest and might help separate out the effects of melanocortins directly on the CNS and immune system from those of corticosteroids. A study of progressive MS, SPMS and/or PPMS should also be considered using the same type of metrics given the limited availability of highly effective treatments for PPMS and the lack of currently approved treatments for SPMS.

The issue of whether ACTH and perhaps even the smaller melanocortins can get through the BBB when there is not a major breakdown of the barrier and thus interact directly with MCR on glia and neurons deserves additional attention. The breakdown of the BBB in MS is likely underestimated since we know that using triple dose gadolinium compared to the single dose used for clinical MRI scans reveals many more enhancing lesions [118,119,120]. As pointed out, proteins considerably larger than ACTH and α-MSH do get through the BBB to a limited extent during periods of clinical stability and to a greater extent during relapses accompanied by gadolinium enhancing lesions on MRI scans. It is now well established that in patients with secondary progressive MS (SPMS) and primary progressive MS (PPMS), gadolinium enhancing lesions occur in as many as 13–25% of patients [120,121,122,123,124,125]. It should also be remembered that most studies of gadolinium enhancing lesions in patients with MS employ standard doses of gadolinium and often 1.5 Tesla magnets. It has been shown using triple dose gadolinium infusions and 3 Tesla magnets that routine single dose gadolinium and 1.0 or 1.5 Tesla magnets greatly underestimate the number of lesions with changes in the BBB [126,127]. There are differences in the BBB and the blood-CSF and blood-meningeal barriers and studies of ACTH within the CNS measure CSF levels, not brain/spinal cord or meningeal levels [128,129,130,131,132,133,134,135]. Thus, even with the rapid breakdown of the peptides, the amount of ACTH and α-MSH that is in the brain and spinal cord may be greater than might be suspected. Additional studies to examine this question are clearly needed.

Exosomes are known to cross the blood brain barrier, and the use of exosomes as a form of delivery has been suggested as a strategy for delivery of disease modifying treatments to the CNS [136,137,138] as has delivery employing nanotechnology [139,140,141,142]. In addition, the problem of breakdown of smaller melanocortins like α-MSH has been approached by modifying the peptide as with NDP-MSH [143], allowing the MSH to better stimulate MCR in vivo. Finally, development of non-peptide ligands for the different MCR that would readily cross the BBB would also have the potential to mediate protection and repair within the CNS in MS and other neurodegenerative diseases.

## 11. Conclusions

It is clear that signaling by melanocortins through their receptors in the CNS has potential for neuroprotection and repair in diseases like MS. This concept is reinforced by our published results showing that the MCR agonist ACTH protects OL, OPC and neurons from excitotoxic, apoptotic, oxidative and inflammation-related effects likely to play a role in CNS damage in MS and other neurodegenerative diseases. While effects on the immune system by direct effects on the circulating cells (lymphocytes and monocytes) and by signaling through CNS cells in regions lacking a mature BBB are clear, additional studies are needed to develop highly effective therapies that directly affect endogenous cells of the CNS, particularly OL, OPC and neurons.

## Figures and Tables

**Figure 1 brainsci-07-00104-f001:**
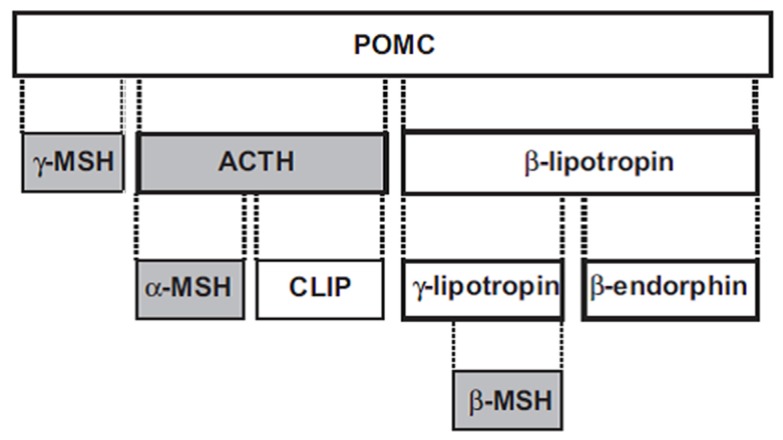
Melanocortin peptides (shaded boxes) derived from POMC. ACTH: adrenocorticotropic hormone; CLIP: corticotropin-like intermediate lobe peptide; MSH: melanocyte-stimulating hormone; POMC: proopiomelanocortin, Reprinted by permission from [8].

**Figure 2 brainsci-07-00104-f002:**
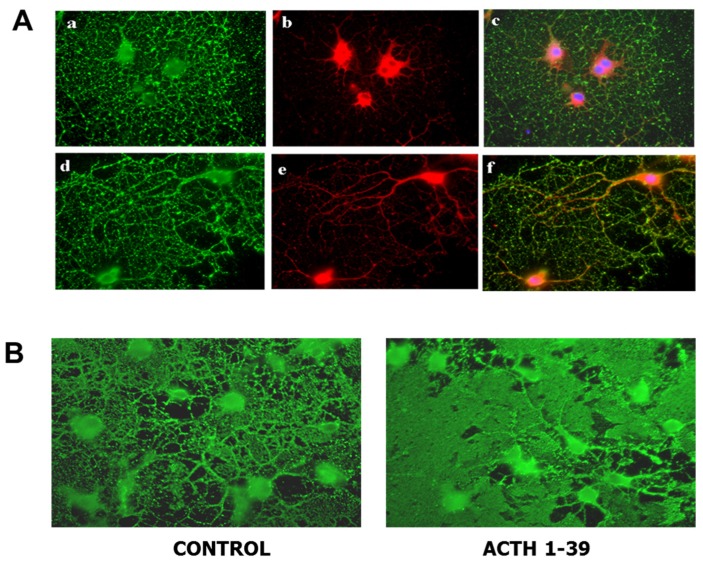
Oligodendroglia express melanocortin receptors and produce larger membrane sheets in response to ACTH. (**A**) Mixed glial cultures from rat brain were immunostained with antibody for MC4R (red) before (**a**–**c**) and after (**d**–**f**) permeabilization to visualize surface and total MC4R respectively; differentiated oligodendroglia were immunostained for surface galactolipid with antibody A007 (green) [96]. (**B**) Oligodendroglia were treated with 200 nM ACTH 1-39 for 3 days, then immunostained with O1 antibody to detect surface galactolipids; ACTH induced larger, more dense membrane sheets [97].

**Table 1 brainsci-07-00104-t001:** ACTH Protects Oligodendroglia, Oligodendroglial Progenitors and Neurons from Multiple Toxic Agents.

Toxic Agent	OL	OPC	Neurons
Glutamate	+	+	+
Staurosporine	+	+	+
Quinolinic acid	+	+	+
Kynurenic acid	none	none	none
H_2_O_2_ (reactive oxygen species	+	+	+
Nitric oxide (slow release)	none	slight	none
Nitric oxide (rapid release)	none	none	slight

ACTH at 200 nM protects cultured rat oligodendroglia (OL), oligodendroglial progenitors (OPC), and neurons from excitotoxic, apoptotic and inflammatory insults, as well as from reactive oxygen species [65,96,97]. No protection was found against kynurenic acid or nitric oxide, except for modest protection for OPC (slow release NO), and for neurons (rapid release NO). Cells were treated for 24 h with the toxic agents in the absence or presence of 200 nM ACTH, the concentration shown to cause maximal protection in these cultures. +, ACTH significantly protected cells from death induced by the toxic agents.

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
