# Peer review of "Melanocortins, Melanocortin Receptors and Multiple Sclerosis"

_brainsci, 2017, doi:10.3390/brainsci7080104_

Round 1

Reviewer 1 Report

Lisak and Benjmins present a very extensive review on the actions of melanocortins in the nervous system and associated diseases.

However, the structure of the manuscript should be strongly revised as the current version results difficult for the reader to follow:

- The revision starts with a generic part on MCRs. But section 3 (Melanocortin receptors) gets too deeply explaining specific actions of peptides and signalling (before explaining the basics on MCR signalling on section 4). The reader expects here a more generic overview on MCRs rather than specific actions on specific cells. The specific part of this section should be explained later included in the sections where the effects of MCs are explained.

-Signaling section (section 4): the current trend on GPCR biology (including MCRs) is to explore novel ways of activating the receptors beyond the classical linear activation, as these novel forms present therapeutic advantages. These include activation with allosteric modulators or biased agonists, both strategies described for MCRs. A comment son these new ways of activating MCRs should be included.

- Section 6: there is an unexplained subdivision in this section (6a) with no further subdivisions. It also looks reasonable that section 7 should be section 6b? As mentioned earlier, the overall structure of the manuscript is difficult to follow.

- Is section 8 necessary? It doesn’t really add anything new as the effects on microglia and astrocytes have already been discussed.

- A key message of the present review is to address the non-cortisol actions of MC peptides. There is a crucial publication (Getting SJ, Arthritis and Rheumatism 2002;46(10):2765-75) that demonstrated this concept by demonstrating the anti-inflammatory effect of ACTH using adrenalectomized rats. This should be included in section 5b.

Other comments:

- A key message of the present review is to address the non-cortisol actions of MC peptides. There is a crucial publication (Getting SJ, Arthritis and Rheumatism 2002;46(10):2765-75) that demonstrated this concept by demonstrating the anti-inflammatory effect of ACTH using adrenalectomized rats. This should be included in section 5b.

- Figure 1: nearly all reviews on MCRs include this same figure. I suggest the authors to present a new figure that might benefit the work and improve reader understanding, showing a diagram or cartoon explaining the model for the anti-inflammatory actions of MCRs in the nervous system (which cells participate, which receptors are involved, which are the actions...etc). This will add novelty to the present manuscript in comparison with previous works.

- One of the limitations of the use of MC peptides is their short half-life. The authors may want to comment on this and the relation this may have with the difficulty in detecting the molecules in CNS.

- There is a recent publication on the effect of melanocortins on EAE, suggesting a role for MC1R by the use of Mc1re/e mice. This work should be commented.

Author Response

Reviewer 1

Lisak and Benjmins present a very extensive review on the actions of melanocortins in the nervous system and associated diseases.

However, the structure of the manuscript should be strongly revised as the current version results difficult for the reader to follow:

- The revision starts with a generic part on MCRs. But section 3 (Melanocortin receptors) gets too deeply explaining specific actions of peptides and signalling (before explaining the basics on MCR signalling on section 4). The reader expects here a more generic overview on MCRs rather than specific actions on specific cells. The specific part of this section should be explained later included in the sections where the effects of MCs are explained.

** We agree that section 3 should be revised as suggested by the reviewer.  We have moved lines 59-97 from section 3 to Section 6 (Direct Effects in the CNS), and reorganized Section 6 as described below.

-Signaling section (section 4): the current trend on GPCR biology (including MCRs) is to explore novel ways of activating the receptors beyond the classical linear activation, as these novel forms present therapeutic advantages. These include activation with allosteric modulators or biased agonists, both strategies described for MCRs. A comment son these new ways of activating MCRs should be included.

** We have added wording and references about allosteric modulators and biased agonists at the end of the first paragraph in section 4.

- Section 6: there is an unexplained subdivision in this section (6a) with no further subdivisions. It also looks reasonable that section 7 should be section 6b? As mentioned earlier, the overall structure of the manuscript is difficult to follow.

** We have reorganized section 6 with additional subheadings to include the information in section 7, as well as the paragraphs moved from section 3.  Sections have been renumbered accordingly.

- Is section 8 necessary? It doesn’t really add anything new as the effects on microglia and astrocytes have already been discussed.

** We agree section 8 is redundant and have removed it.

- A key message of the present review is to address the non-cortisol actions of MC peptides. There is a crucial publication (Getting SJ, Arthritis and Rheumatism 2002;46(10):2765-75) that demonstrated this concept by demonstrating the anti-inflammatory effect of ACTH using adrenalectomized rats. This should be included in section 5b.

**  We thank the reviewer for noting the paper by Getting et al. We have added the information and reference to section 5b.  

Other comments:

- A key message of the present review is to address the non-cortisol actions of MC peptides. There is a crucial publication (Getting SJ, Arthritis and Rheumatism 2002;46(10):2765-75) that demonstrated this concept by demonstrating the anti-inflammatory effect of ACTH using adrenalectomized rats. This should be included in section 5b.

** This reference is now included, as noted above.

- Figure 1: nearly all reviews on MCRs include this same figure. I suggest the authors to present a new figure that might benefit the work and improve reader understanding, showing a diagram or cartoon explaining the model for the anti-inflammatory actions of MCRs in the nervous system (which cells participate, which receptors are involved, which are the actions...etc). This will add novelty to the present manuscript in comparison with previous works.

** We have elected to retain Figure 1 since it is useful as an introduction to readers not familiar with melanocortins. Several recent reviews have summarized the anti-inflammatory actions of MCRs in the nervous system, while the novel focus of our review is on the roles of MCR in neuroprotection through direct actions on oligodendroglia separate from anti-inflammatory effects.

- One of the limitations of the use of MC peptides is their short half-life. The authors may want to comment on this and the relation this may have with the difficulty in detecting the molecules in CNS.

**We have noted the short half-lives of the MC peptides with regard to CNS levels at the end of the 4th paragraph of Section 10 (p. 23).

- There is a recent publication on the effect of melanocortins on EAE, suggesting a role for MC1R by the use of Mc1re/e mice. This work should be commented.

We have added comments about the publication suggesting a role for MC1R in EAE (Mykicki et al.) to section 7.

Reviewer 2 Report

Overall this is an interesting review and comprehensively covers roles for the melanocortin system in the central and peripheral nervous systems.  It is also an important review because the melanocortins are studied a great deal due to the important roles they play in metabolism.  There are fewer studies on the broader roles they play in the peripheral and central nervous systems where there is potential to use these peptides in various therapies.

Specific Points:

1. The title focuses on Multiple Sclerosis. However, I suggest that this be revised since the review is more broadly focused on neural protective and neural repair mechanisms together with treatments for neurologic diseases.

2. There are sections in this review which would benefit from editing.  There are many long sentences with poor punctuation and poor grammar e.g lines 20-24, lines 375-377, lines 420-423.

3. Sections numbers need revising.  There is section 6a followed by section 7.  It appears that section 7 should be 6b.

4. Table 1 show effects of exogenous ACTH on cultured neurons. This is using a pharmacological dose and the time of exposure is not mentioned. While this data is interesting it may not reflect in vivo physiological actions of ACTH. Is 200nM ACTH required to see these effects or were lower doses of ACTH also responsive?

5. Sections 8 and 9 start with 'As noted earlier' and 'As described earlier". It would helpful if the authors could refer to the sections where these were described earlier.

6. While the first paragraph of section 10 is interesting, there is no consensus here stating what point is being made. Information in the first paragraph is picked up again in lines 374-375 in the middle of the second paragraph.  I think this could be revised so it is easier to read.

7. Author contributions and conflicts of interest are not completed.

Author Response

Reviewer 2

Comments and Suggestions for Authors

Overall this is an interesting review and comprehensively covers roles for the melanocortin system in the central and peripheral nervous systems.  It is also an important review because the melanocortins are studied a great deal due to the important roles they play in metabolism.  There are fewer studies on the broader roles they play in the peripheral and central nervous systems where there is potential to use these peptides in various therapies.

Specific Points:

1. The title focuses on Multiple Sclerosis. However, I suggest that this be revised since the review is more broadly focused on neural protective and neural repair mechanisms together with treatments for neurologic diseases.

**We have retained the title since purpose of the review and our studies is to identify therapeutic targets for the treatment of MS.  A large portion of section 9 (Treatment of Neurologic Diseases with Melanocortins) discusses multiple sclerosis.

2. There are sections in this review which would benefit from editing.  There are many long sentences with poor punctuation and poor grammar e.g lines 20-24, lines 375-377, lines 420-423.

** We have rewritten the sentences noted to improve clarity and grammar.

3. Sections numbers need revising.  There is section 6a followed by section 7.  It appears that section 7 should be 6b.

** Section 6 has been reorganized and expanded, as described in our responses to Reviewer 1.. The section now has three parts (a, b, c),

4. Table 1 show effects of exogenous ACTH on cultured neurons. This is using a pharmacological dose and the time of exposure is not mentioned. While this data is interesting it may not reflect in vivo physiological actions of ACTH. Is 200nM ACTH required to see these effects or were lower doses of ACTH also responsive?

**The legend to Table 1 now contains the requested information.

5. Sections 8 and 9 start with 'As noted earlier' and 'As described earlier". It would helpful if the authors could refer to the sections where these were described earlier.

** We now refer to specific sections as requested.

6. While the first paragraph of section 10 is interesting, there is no consensus here stating what point is being made. Information in the first paragraph is picked up again in lines 374-375 in the middle of the second paragraph.  I think this could be revised so it is easier to read.

**We have revised as requested for clarity.

7. Author contributions and conflicts of interest are not completed.

** These have been added
